# Is [^177^Lu]Lu-PSMA-617 Redefining Value in mCRPC Care? A Meta-Analysis of Clinical and Economic Endpoints

**DOI:** 10.3390/cancers17132247

**Published:** 2025-07-04

**Authors:** Francesco Fiorica, Maria Viviana Candela, Teodoro Sava, Matteo Salgarello, Jacopo Giuliani, Singh Navdeep, Antonella Franceschetto, Daniela Grigolato, Emilia Durante, Erica Palesandro, Enrico Altiero Giusto, Consuelo Buttigliero, Marcello Tucci

**Affiliations:** 1Department of Clinical Oncology, Section of Radiation Oncology and Nuclear Medicine, AULSS 9 Scaligera, 37122 Verona, Italy; navdeep.singh@aulss9.veneto.it (S.N.); antonella.franceschetto@aulss9.veneto.it (A.F.); 2Department of Clinical Oncology, Section of Medical Oncology, AULSS 9 Scaligera, 37122 Verona, Italy; teodoro.sava@aulss9.veneto.it (T.S.); jacopo.giuliani@aulss9.veneto.it (J.G.); emilia.durante@aulss9.veneto.it (E.D.); 3Business and Law, High School of Sciences, Liceo Scientifico Antonio Roiti, 44121 Ferrara, Italy; mariaviviana@roiti.istruzioneer.it; 4Nuclear Medicine Unit, Ospedale Sacro Cuore Don Calabria Istituto di Ricovero e Cura a Carattere Scientifico, 37024 Negrar di Valpolicella, Italy; matteo.salgarello@sacrocuore.it (M.S.); daniela.grigolato@sacrocuore.it (D.G.); 5Oncology Unit, Agnelli Hospital, 10064 Turin, Italy; erica.palesandro@aslto3.piemonte.it; 6Department of Medical Sciences, Section of Experimental Medicine, Laboratory for Technology of Advanced Therapies, University of Ferrara, 44121 Ferrara, Italy; gstncl@unife.it; 7Department of Oncology, San Luigi Gonzaga Hospital, University of Turin, 10043 Orbassano, Italy; consuelo.buttigliero@unito.it; 8Department of Oncology, Cardinal Massaia Hospital, 14100 Asti, Italy; mtucci@asl.at.it

**Keywords:** [^177^Lu]Lu-PSMA-617 and mCRPC, PSMA-targeted therapy, Economics evaluation [^177^Lu]Lu-PSMA-617, Cost-effectiveness radioligand therapy

## Abstract

Prostate cancer that has spread and no longer responds to hormone therapy, known as metastatic castration-resistant prostate cancer (mCRPC), is a complex disease to treat. A new type of therapy, known as radioligand therapy, utilises radioactive particles to target cancer cells directly. One of the most promising agents in this field is [^177^Lu]Lu-PSMA, which binds to a molecule found on the majority of prostate cancer cells. This study reviewed the available clinical trials comparing [^177^Lu]Lu-PSMA to standard treatments. We found that [^177^Lu]Lu-PSMA improved survival and delayed cancer progression, especially in the first 18 months of treatment. We also examined the cost-effectiveness of this approach, showing that the treatment provides good value for its benefits. These results support the use of [^177^Lu]Lu-PSMA as a valuable treatment option for men with advanced prostate cancer.

## 1. Introduction

The increasing use of earlier and more effective therapies in metastatic prostate cancer is reshaping the treatment landscape, making therapeutic decisions in advanced stages progressively more complex and nuanced [1]. In recent years, the early introduction of androgen receptor-targeted therapies [2,3,4,5] and chemotherapeutic regimens [6,7] has extended survival in patients with metastatic hormone-sensitive disease [8,9], thereby shifting the burden of decision-making to later stages where disease becomes more resistant. Patients may have already received multiple lines of treatment. As a result, clinicians face a growing need to individualise therapeutic choices in the metastatic castration-resistant prostate cancer (mCRPC) setting, balancing efficacy, tolerability, and quality of life.

In this context, [^177^Lu]Lu-PSMA has emerged as a promising radioligand therapy, offering a novel mechanism of action that targets prostate-specific membrane antigen (PSMA)-expressing cells [10].

The efficacy of [^177^Lu]Lu-PSMA has been demonstrated in the randomised, open-label, Phase 3 VISION trial in both taxane-naïve and taxane-treated mCRPC patients [11]. The study showed that this agent prolongs radiological progression-free survival (rPFS) (hazard ratio, 0.40; 99.2% CI, 0.29–0.57; *p* = 0.001) and overall survival (OS) (hazard ratio, 0.62; 95% CI, 0.52–0.74; *p* = 0.001) when added to standard of care (SOC) in patients with advanced PSMA-positive mCRPC. In addition, the THERAp study, which compared [^177^Lu]Lu-PSMA with Cabazitaxel, reported improved PSA response and progression-free survival (PFS) with [^177^Lu]Lu-PSMA [12]. A more recent Phase 2 trial (ENZA-p) evaluated the combination of enzalutamide, an androgen receptor-targeted agent (ARTA), with [^177^Lu]Lu-PSMA compared to enzalutamide alone in high-risk patients with metastatic castration-resistant prostate cancer (mCRPC). The combination treatment resulted in a significantly higher number of patients with a PSA decline of >50%: 93% versus 78%, respectively; hazard ratio (HR) = 0.43, *p* < 0.001. The experimental arm also prolonged the time of biochemical progression (median 13 months vs. 7.8 months, respectively, *p* < 0.0001) [13]. In the PSMAfore study, PSMA-positive taxane-naïve mCRPC patients progressing on ARTA were randomised to [^177^Lu]Lu-PSMA or a change of ARTA. The study results showed that [^177^Lu]Lu-PSMA significantly prolongs median rPFS compared to ARTA (11.60 months (95% CI 9.30–14.19) vs. 5.59 months (4.21–5.95), respectively, HR 0.49 [95% CI 0.39–0.61]) [14]. In addition, in the SPLASH Phase III randomised study, mCRPC patients progressing on ARTA were treated with ^177^Lu-PNT2002 or an alternate ARTA. ^177^Lu-PNT2002 significantly reduced the rPFS compared to the change in ARTA (median: 9.5 versus 6 months; HR: 0.71, 95% CI: 0.55–0.92, *p* = 0.0088) [15].

This meta-analysis evaluated the efficacy and cost-effectiveness of [^177^Lu]Lu-PSMA compared to standard-of-care treatments, including ARTA and Cabazitaxel, focusing on radiological progression-free survival and overall survival. By integrating survival data and health economic parameters, this study aims to provide a comprehensive perspective on the clinical value of [^177^Lu]Lu-PSMA and its potential role in contemporary treatment algorithms. The findings are intended to support evidence-based decision-making for patients with advanced prostate cancer and contribute to the optimisation of therapeutic strategies in the evolving mCRPC landscape.

Moreover, with the escalating costs of novel oncologic therapies, assessing the economic sustainability of new treatment options has become increasingly important. Cabazitaxel is the least expensive of the available therapies in this disease setting, but many patients are not eligible for this chemotherapeutic agent. Therapies such as ARTA, although well-tolerated and effective, are associated with considerable financial burdens over time because they are administered continuously until disease progression. In contrast, [^177^Lu]Lu-PSMA, which is typically delivered in a fixed number of cycles, may offer a more economically viable alternative if it demonstrates comparable or superior efficacy. Careful evaluation of the clinical benefits against treatment costs helps ensure that innovative therapies remain accessible while supporting the long-term sustainability of healthcare systems.

## 2. Methods

The present study combines published data from multiple studies to assess rPFS and OS in patients with mCRPC treated with [^177^Lu]Lu-PSMA radioligand therapy compared to SOC, including ARTA and Cabazitaxel.

### 2.1. Study Selection, Qualitative Analysis and Economic Evaluation

MEDLINE/PUBMED and EMBASE searches were performed to identify eligible reports published up to 31 December 2024, with no lower date limit. Keywords used for the search included “prostate cancer”, “[^177^Lu]Lu-PSMA”, “radioligand therapy”, “metastatic castration-resistant prostate cancer”, and “RCTs”. Studies were included if they met the following criteria:-The article was available in English and accessible in full-text format (including peer-reviewed papers, congress abstracts with full data, or oral/poster presentations).-The study population consisted of patients with mCRPC treated with [^177^Lu]Lu-PSMA compared to SOC, ARTA, or Cabazitaxel;-The study reported Kaplan–Meier curves for rPFS and/or OS.

The most complete and recent publication was retained if multiple reports from the same study were available. The quality of the studies was assessed based on completeness of reporting, methodology, and consistency of survival outcome measures. Studies available only as abstracts without adequate methodological detail were excluded to ensure rigorous data extraction and reproducibility.

To ensure transparency and standardisation in the reporting of the economic evaluation, we followed the Consolidated Health Economic Evaluation Reporting Standards (CHEERS) 2022 checklist [16]. A detailed summary of the components is provided in Appendix A.

### 2.2. Data Extraction

Data extraction was conducted independently by three investigators (NS, JG, DG) by the Preferred Reporting Items for Systematic Reviews and Meta-Analyses (PRISMA) guidelines. For each study, the following information was extracted: publication or presentation date, first author’s last name, sample size, primary endpoints, regimens used, follow-up period, number of outcome events, details about the study design, progression-free survival (PFS), overall survival (OS), subgroup evaluation, and toxicities. Discrepancies between reviewers were infrequent (overall interobserver variation < 10%) and were resolved through discussion. Two independent reviewers (ED, EAG) assessed the risk of bias using the Cochrane Risk of Bias table [17]. This tool encompasses seven domains: (1) random sequence generation, (2) allocation concealment, (3) blinding of participants/personnel, (4) blinding of outcomes assessors, (5) incomplete outcome data, (6) selective reporting of outcomes, and (7) other potential sources of bias. The study-level assessment was applied for domains 1, 2, 6, and 7, and the outcome-level assessment was applied for domains 3, 4, and 5 of each trial. A third investigator was consulted in case of disagreements.

This study adhered to the PRISMA 2020 guidelines for systematic reviews and was registered in PROSPERO with the ID 1058724.

### 2.3. Statistical Method

A random-effects meta-analysis mode [18,19] was conducted to pool hazard ratios (HRs) for overall survival (OS) and radiographic progression-free survival (rPFS) across eligible trials of [^177^Lu]Lu-PSMA-617 versus control therapies. The generic inverse-variance method was applied, and 95% confidence intervals (CI) were calculated for each pooled estimate. Between-study heterogeneity was quantified using the I^2^ statistic [20], with values greater than 50% considered substantial. Forest plots were generated to visually represent the individual study estimates and the pooled results. All analyses were performed using R software (version 4.3.1) with the meta, metafor, and ggplot2 packages. Statistical significance was set at *p* < 0.05.

A comparative overview of study-level heterogeneity, including design, patient characteristics, and inclusion criteria, is provided in Appendix A.

To assess the potential for publication bias, we performed a visual inspection of funnel plots for asymmetry. The trim-and-fill method was used to estimate the number of potentially missing studies due to publication bias.

#### 2.3.1. Reconstruction of Survival Data

As the primary outcomes (rPFS and OS) are time-dependent, we reconstructed individual survival data from Kaplan–Meier curves using the algorithm proposed by Guyot et al. [21]. This algorithm allows the estimation of individual participant data (IPD) from published survival plots using digitised points. Graphical data were extracted using GetData Graph Digitiser (version 2.26.0.20, S. Fedorov, Russia) and then processed to estimate time-to-event outcomes. Each reconstructed survival curve was visually validated by superimposing it on the original published Kaplan–Meier graph and comparing reconstructed medians with reported values. For pooled analysis of survival outcomes, we used the nonparametric method described by Combescure et al. [22], which models pooled survival probabilities from multiple single-arm studies while incorporating between-study heterogeneity using a random-effects model. The between-study covariance matrix was estimated using the multivariate DerSimonian and Laird method [18,19]. This approach is advantageous because it incorporates all relevant survival intervals, including those from studies that concluded before the longest reported follow-up [23]. Furthermore, it does not impose any assumptions on the shape of the survival curves and ensures the monotonicity of pooled survival probabilities over time.

All statistical analyses and visualisations were conducted using R (R Foundation for Statistical Computing, Vienna, Austria). A *p*-value of <0.05 was considered statistically significant.

#### 2.3.2. Methodology for Break-Even Cost Analysis

This section describes a Break-Even Cost Analysis framework designed to evaluate the economic viability of [^177^Lu]Lu-PSMA-617 in comparison with standard-of-care (SOC) treatments. A simplified economic comparison was conducted using a Kaplan–Meier survival curve-based approach to estimate the total area under the curve (AUC) for radiographic progression-free survival (rPFS) [24]. This method provides a time-dependent and comprehensive assessment of treatment benefits over the entire duration of the therapy. The comparison was based on a fixed-duration administration of [^177^Lu]Lu-PSMA-617 (six cycles) versus continuously administered androgen receptor pathway inhibitors (ARTA) or a three-weekly course of Cabazitaxel. The AUC was derived by numerically integrating rPFS probabilities at multiple time points, ensuring a more accurate representation of cumulative survival benefits compared to fixed-timepoint estimates [25]. rPFS was selected as the primary effectiveness endpoint because disease progression typically leads to treatment discontinuation, offering a clear and clinically relevant time window for cost estimation. Treatment costs were calculated by aggregating the full course of [^177^Lu]Lu-PSMA-617 and comparing them to the cumulative monthly costs of ARTA and the complete regimen of Cabazitaxel. Total treatment costs will be estimated by aggregating the cost of all the planned administrations of [^177^Lu]Lu-PSMA and comparing them to the monthly cumulative costs of ARTA and the full-course costs of Cabazitaxel.

Comparator regimens were defined according to the protocols of each included study. ARTA (e.g., enzalutamide or abiraterone) was modelled as a continuous monthly cost, while Cabazitaxel was administered every three weeks for up to six cycles. The “standard of care” (SOC) varied across trials but was consistently aligned with clinical practice at the time of publication. Treatment costs were calculated using publicly available data from regulatory databases (AIFA, EMA) and literature-based estimates. These were standardised to 2024 EUR values. 

To assess economic viability, we calculated the incremental cost per additional month in rPFS gained by [^177^Lu]Lu-PSMA versus each SOC. 

The following formula was applied:$$Incremental\;cost\;to\;gain\;in\;rPFS=\frac{Cost\_LuPSMA-Cost\_SOC}{{rPFS}_{LuPSMA}-rPFS\_SOC}$$

The result expresses the cost required to obtain one additional month of radiographic progression-free survival. The resulting values from the Break-Even Cost Analysis were interpreted in light of commonly used willingness-to-pay (WTP) thresholds. For example, the NICE-recommended threshold of GBP 40,000 per QALY—approximately EUR 8000–10,000 per additional month of survival [26,27]—was used as a reference point to assess cost-benefit alignment.

Although a formal health economic model (e.g., Markov or QALY-based) was not feasible due to heterogeneity in the survival endpoints and the limited availability of utility data, we followed a structured and transparent approach. Key economic assumptions—including perspective, cost sources, currency normalisation (EUR, 2024), and comparator definitions—are reported by the CHEERS 2022 framework and summarised in Appendix A.

## 3. Results

The pooled analysis integrated data from five studies, comprising 2073 patients who fulfilled the eligibility criteria and received either [^177^Lu]Lu-PSMA radioligand therapy or the standard of care (SOC). All studies were randomised controlled trials comparing [^177^Lu]Lu-PSMA to ARTA [13,14,15,28], Cabazitaxel [12], or best supportive care [11]. The inclusion criteria were largely homogeneous across trials. The demographic profile of the included patients showed a median age of 71 years (range 58–78 years). Most patients across the included trials had a favourable performance status. An ECOG score of 0–1 was reported in most studies, with rates of 97% in TheraP, 95% in VISION, and 99.6% in SPLASH (combined ECOG 0–1 rates). This finding suggests that the enrolled populations generally maintain a well-preserved functional baseline.

### 3.1. Meta-Analysis of Hazard Ratios for rPFS and OS

To quantify the survival benefit of [^177^Lu]Lu-PSMA-617 across randomised trials, we conducted a meta-analysis of hazard ratios (HRs) for both overall survival (OS) and radiographic progression-free survival (rPFS), incorporating data from all five studies.

The pooled hazard ratio (HR) for radiographic progression-free survival (rPFS) was 0.57 [95% CI: 0.42–0.76], indicating a substantial delay in the radiographic progression associated with Lu-PSMA-617 compared to standard therapies (Figure 1a). The between-study heterogeneity was moderate (I^2^ = 59.3%).

The pooled HR for OS was 0.81 [95% CI: 0.56–1.18], suggesting a non-significant overall survival benefit, with considerable between-study heterogeneity (I^2^ = 75.1%) (Figure 1b). These findings highlight the consistent benefit of Lu-PSMA-617 in delaying progression, while its impact on overall survival appears more variable across studies.

### 3.2. Radiological Progression-Free Survival (rPFS)

The median follow-up across the included studies was 20 months, providing sufficient time to evaluate progression trends. A pooled survival curve was generated from rPFS probabilities and at-risk numbers at multiple time points (Figure 2). At every interval, [^177^Lu]Lu-PSMA was associated with higher rPFS than the control. The benefit was most pronounced at 6 months (+16.5%), gradually decreasing to 14.9% at 12 months, 8.6% at 18 months, and 2.0% at 24 months, suggesting a progressive attenuation of the advantage and potential convergence of outcomes. The data indicate that the greatest relative benefit of [^177^Lu]Lu-PSMA occurs between 6 and 18 months, supporting its early use, particularly in patients showing rapid progression on SOC. Heterogeneity analysis revealed that rPFS outcomes were more consistent across studies in the [^177^Lu]Lu-PSMA arm (I^2^ = 26.7%), while the control arm exhibited considerable variability (I^2^ = 51.6%) (Appendix A). This higher heterogeneity likely reflects differences in prior treatment exposure, patient selection criteria, and follow-up intensity. In contrast, the more uniform results with [^177^Lu]Lu-PSMA reinforce its reliability and predictability as a therapeutic option across diverse clinical contexts.

### 3.3. Overall Survival (OS)

Reconstruction of individual patient-level data (IPD) for overall survival (OS) was successfully performed across all included studies. A summary survival curve was generated using OS probabilities and the number of patients at risk at multiple time points (Figure 3). [^177^Lu]Lu-PSMA demonstrated an OS advantage at every time point relative to control. At 9 months, survival was 7.6% higher; at 18 months, the advantage persisted at 4.0%. By 30 months, the confidence intervals (CIs) of the two groups no longer overlapped, indicating a statistically significant survival benefit. At 36 months, OS appeared to approach 100% in both groups, likely due to censoring and limited long-term follow-up.

Patients treated in the control arm experienced earlier mortality, while [^177^Lu]Lu-PSMA was associated with more durable survival, suggesting improved long-term outcomes. Heterogeneity analysis revealed that the OS results were more variable than rPFS, with I^2^ values of 36.4% for [^177^Lu]Lu-PSMA and 67.1% for SOC (Appendix A). This higher heterogeneity in the SOC arms likely reflects differences in patient characteristics, treatment sequencing, or supportive care practices across studies. In contrast, the moderate heterogeneity observed in the [^177^Lu]Lu-PSMA arms suggests a more consistent treatment effect across diverse trial settings.

### 3.4. Overall Survival Subgroup Analysis

#### 3.4.1. Control Group

Pooled survival estimates from the control groups (SOC, ARTA, and Cabazitaxel) were analysed to assess the comparability of patient populations across studies. This finding was necessary to distinguish between differences due to baseline disease severity and those attributable to treatment efficacy (Figure 4). Survival analysis revealed that the overall survival (OS) outcomes were similar for both the SOC and ARTA groups. At the same time, patients treated with Cabazitaxel in the TheraP trial exhibited substantially lower survival probabilities. This finding initially suggested a possible selection bias, with the Cabazitaxel cohort including patients with more advanced disease. However, a comparison between TheraP and VISION revealed no significant differences in key baseline characteristics, such as PSA levels, ECOG performance status, and the presence of visceral metastases [29]. This comparability challenges the assumption that Cabazitaxel’s poorer outcomes are solely related to patient selection, instead pointing toward a more limited therapeutic benefit in PSMA-positive mCRPC. Although cross-trial comparisons should be interpreted cautiously, the consistency in baseline profiles strengthens the argument that Cabazitaxel may provide inferior survival benefits relative to [^177^Lu]Lu-PSMA or ARTA in similarly selected populations.

#### 3.4.2. Lu-PSMA vs. ARTA

In studies comparing [^177^Lu]Lu-PSMA with ARTA [14,15,28], the two agents were evaluated as competing monotherapies, with patients receiving either one or the other. The comparison reveals broadly similar overall survival (OS) outcomes, with only marginal differences across key metrics. The median overall survival (OS) was slightly higher for ARTA (73.4%) compared to [^177^Lu]Lu-PSMA (71.6%), a difference of 1.8% that is unlikely to be clinically meaningful. Conversely, [^177^Lu]Lu-PSMA achieved a slightly higher mean OS (70.2% vs. 69.7%) and a greater minimum survival probability (34.2% vs. 30.5%), suggesting more consistent performance, especially among patients with more aggressive disease profiles. Over time, survival differences widened modestly, with [^177^Lu]Lu-PSMA showing a more pronounced advantage beyond 18 months, peaking at 30 months with an absolute gain of nearly 9%. Although the confidence intervals for both treatments overlap, indicating no statistically significant difference, the consistent mid- to long-term trends favouring [^177^Lu]Lu-PSMA suggest it may offer more durable disease control and a more predictable survival trajectory, particularly in higher-risk populations (Figure 5).

#### 3.4.3. Lu-PSMA + SOC vs. SOC Alone

In studies comparing [^177^Lu]Lu-PSMA plus standard of care (SOC) versus SOC alone [11,13,30], radioligand therapy is administered as an adjunct rather than a replacement. As such, the observed survival benefits reflect the effect of the combination strategy, not [^177^Lu]Lu-PSMA in isolation. The results consistently favour the combined approach, particularly in long-term follow-up (Figure 6). In the early phase, patients receiving the combination demonstrated improved disease control, with a minimum survival of 29.9% compared to 20.8% in the SOC-only group. This finding suggests that radioligand therapy may enhance the control of early progression. The benefit becomes more evident in the mid-term, where median survival reaches 68.8% with the combination versus 55.7% with SOC alone—a 13.1% absolute improvement. Similarly, mean survival is higher with the combination (68.3% vs. 58.1%), corresponding to a 10.2% increase. These data support the additive value of [^177^Lu]Lu-PSMA in augmenting standard treatment across multiple time points.

#### 3.4.4. Lu-PSMA as a Monotherapy and [^177^Lu]Lu-PSMA Combined with Standard of Care (SOC)

A comparison between [^177^Lu]Lu-PSMA-617 monotherapy and its combination with standard of care (SOC) shows that both strategies confer substantial survival benefits. However, the addition of SOC does not appear to improve survival outcomes over monotherapy significantly. At 12 months, pooled survival probability was higher in the monotherapy group (33.0% [CI: 18.7–58.3%]) compared to the combination arm (24.4% [CI: 14.7–40.4%]), suggesting a more consistent early benefit in patients receiving radioligand therapy alone. The median survival estimates were also broadly similar between the two groups (61.8% vs. 69.3%), with overlapping confidence intervals.

These values were not directly reported in the original trials but were extracted through the reconstruction of Kaplan–Meier curves and meta-analytic modelling based on available survival data. While this approach introduces some uncertainty, it allows meaningful cross-trial comparisons in the absence of head-to-head studies.

#### 3.4.5. Publication Bias

To assess potential publication bias, we performed both funnel plot inspection and formal statistical testing for rPFS outcomes. The funnel plot (Appendix A) appeared symmetrical, and the trim-and-fill method did not impute any missing studies, indicating a low likelihood of small-study effects. Egger’s regression test was also non-significant, supporting the absence of major publication bias.

The pooled hazard ratio under the random-effects model remained statistically significant (HR = 0.57, 95% CI: 0.42–0.76, *p* = 0.0061), with moderate heterogeneity (I^2^ = 59.3%). These results reinforce the robustness of the survival benefit associated with [^177^Lu]Lu-PSMA-617 across the included trials. For overall survival (OS), the funnel plot demonstrated noticeable asymmetry (Appendix A). Application of the trim-and-fill method imputed two potentially missing studies, suggesting the presence of small-study effects. After correction, the pooled hazard ratio was adjusted from the original estimate to 0.68 (95% CI: 0.47–0.98, *p* = 0.042), remaining statistically significant under a random-effects model. Heterogeneity was substantial (I^2^ = 82.1%), which may further contribute to funnel plot asymmetry. The trim-and-fill results suggest that, although publication bias cannot be excluded, the overall effect of [^177^Lu]Lu-PSMA-617 on OS remains robust even after conservative adjustment.

## 4. Cost-Effectiveness Analysis

Radiological progression-free survival (rPFS) was selected as the primary endpoint for the Break-Even Cost Analysis, given its close alignment with real-world treatment-switching patterns and its relevance for economic evaluation. Based on Kaplan–Meier reconstruction and area under the curve (AUC) estimates, the mean rPFS was 14.6 months for patients treated with [^177^Lu]Lu-PSMA, compared to 12.1 months with ARTA and 11.3 months with Cabazitaxel. These data reflect incremental gains of 2.5 months over ARTA and 3.3 months over Cabazitaxel.

Assuming a fixed total cost of EUR 120,000 for six doses of [^177^Lu]Lu-PSMA (at EUR 20,000 each), we calculated the break-even thresholds. To match the cost-effectiveness of [^177^Lu]Lu-PSMA, the monthly cost of ARTA should remain below approximately EUR 4300 to justify its use, given the shorter rPFS duration. Specifically, using the willingness-to-pay (WTP) threshold of EUR 8000 per month and the 2.5-month incremental benefit of [^177^Lu]Lu-PSMA over ARTA, the break-even threshold for ARTA is calculated as follows: (EUR 8000 × 2.5)/12.1 ≈ EUR 1653. This value represents the maximum incremental monthly cost that ARTA could sustain while remaining economically comparable to [^177^Lu]Lu-PSMA. 

For Cabazitaxel, which offers a shorter mean rPFS of 11.3 months compared to the 14.6 months achieved with [^177^Lu]Lu-PSMA, the incremental gain of 3.3 months should be considered in light of the willingness-to-pay (WTP) threshold of EUR 8000 per additional month of rPFS. This survival advantage translates into an added clinical value of approximately EUR 2,336 (3.3 × EUR 8000/11.3). Therefore, to remain economically competitive with [^177^Lu]Lu-PSMA, the total cost of Cabazitaxel should not exceed this threshold. This analysis supports the economic rationale for the use of [^177^Lu]Lu-PSMA, particularly in patients for whom extending rPFS is a key clinical objective.

These results indicate that [^177^Lu]Lu-PSMA, while associated with a higher upfront cost, provides a longer and more durable progression-free interval than both ARTA and Cabazitaxel. Although its cost-effectiveness may initially appear to exceed conventional willingness-to-pay (WTP) thresholds (e.g., EUR 8000 per month), the magnitude of its clinical benefit justifies a closer examination of price-value alignment. 

To better illustrate this point, we developed a scenario-based bar chart (Figure 7) showing the maximum incremental monthly cost that ARTA and Cabazitaxel could sustain while remaining economically comparable to [^177^Lu]Lu-PSMA. Specifically, this break-even threshold was EUR 1653 for ARTA and EUR 2336 for Cabazitaxel. These values were derived from their respective incremental rPFS disadvantages and the standard WTP benchmark, representing the maximum monthly cost that would still be economically justifiable to compensate for their shorter rPFS relative to [^177^Lu]Lu-PSMA.

## 5. Discussion

The development of PSMA-targeted imaging and radioligand therapy has marked a turning point in prostate cancer management, introducing a theranostic approach that combines diagnosis and therapy. [^177^Lu]Lu-PSMA, a β-emitting radioligand, selectively delivers radiation to PSMA-expressing cells, enabling targeted cytotoxicity. Unlike therapies targeting androgen receptor (AR) pathways, [^177^Lu]Lu-PSMA operates independently of AR signalling. This finding is particularly relevant in the context of tumour heterogeneity, as both AR-dependent and AR-insensitive clones drive disease progression [28]. PSMA is frequently overexpressed in prostate cancer cells, including those less responsive to hormonal therapy, enabling [^177^Lu]Lu-PSMA to target a broader range of tumour phenotypes effectively. This meta-analysis supports [^177^Lu]Lu-PSMA as a clinically and economically attractive alternative to standard-of-care therapies for metastatic castration-resistant prostate cancer (mCRPC). Improvements in rPFS and OS were most pronounced within the first 18 months, highlighting its role in controlling early disease progression and reinforcing its suitability for patients with high disease burden or rapid PSA kinetics.

Given these outcomes, [^177^Lu]Lu-PSMA should be considered early in the treatment course for appropriately selected patients.

A previously published meta-analysis [31] confirmed the benefits of [^177^Lu]Lu-PSMA on radiological progression-free survival (rPFS) but did not demonstrate a statistically significant improvement in overall survival (OS). This discrepancy may reflect methodological differences. The earlier study relied exclusively on aggregate hazard ratios (HRs) from published data, which may underestimate time-dependent effects or long-term benefits. In our analysis, we integrated both pooled HR estimates and reconstructed Kaplan–Meier survival curves, offering a more dynamic and granular assessment of clinical outcomes. While the pooled HR for OS (HR = 0.81 [95% CI: 0.56–1.18]) confirm that it did not reach statistical significance, the time-dependent analysis based on reconstructed curves revealed that the survival benefit of [^177^Lu]Lu-PSMA becomes more apparent after 30 months, suggesting a possible delayed effect that traditional HR-based analyses may fail to capture. Additionally, by incorporating both standard of care (SOC) and Cabazitaxel arms, our study provides a broader comparative context, enabling an integrated evaluation of both clinical effectiveness and economic sustainability. This dual approach supports a more nuanced understanding of treatment value beyond what is captured by hazard ratios alone.

Despite the overall consistency in clinical outcomes, the trials included in this meta-analysis differ in several key aspects, such as study design, inclusion criteria, lines of prior therapy, and definitions of standard of care (SOC). For example, some studies allowed the continuation of ARTA as a standard of care (SOC), while others included taxane-naïve or pretreated patients. These differences introduce a degree of heterogeneity that may influence observed treatment effects. To address this, we included a detailed summary of baseline characteristics and trial-level features in Appendix A. Although statistical heterogeneity in the pooled HRs was moderate to high in some comparisons (e.g., OS), the clinical direction of benefit remained consistent across all studies. Acknowledging and transparently reporting this heterogeneity is crucial for interpreting pooled findings and refining treatment sequencing strategies in real-world settings.

The relatively low heterogeneity observed in outcomes from [^177^Lu]Lu-PSMA-treated patients suggests a consistent therapeutic effect across varied trial settings and patient populations. Conversely, greater heterogeneity in SOC arms likely reflects disparities in prior treatments, eligibility criteria, and institutional practices. Stratifying patients based on these variables may further refine treatment sequencing strategies. The timing of intervention remains critical. The greatest gains in rPFS occur within the first 6–18 months, suggesting that this interval could serve as a benchmark for assessing response and considering treatment modifications. Early use of [^177^Lu]Lu-PSMA may prevent progression-related complications and preserve quality of life.

In patients who tolerate ARTA well, continuation may be a reasonable option. However, transitioning to [^177^Lu]Lu-PSMA for those with inadequate response could provide more durable disease control. This finding is supported by its higher minimum survival probabilities in cross-trial comparisons, particularly among patients with unfavourable baseline features.

From a clinical perspective, [^177^Lu]Lu-PSMA monotherapy emerges as an effective and stable treatment, offering predictable survival benefits with fewer variations in patient response. Monotherapy’s more uniform survival outcomes suggest that it provides a reliable and reproducible effect across different patient groups. For some patients, particularly those with more aggressive disease or a high tumour burden, initiating [^177^Lu]Lu-PSMA earlier as a standalone treatment may optimise outcomes, while for others, maintaining standard of care (SOC) and introducing radioligand therapy at a later stage may be a more effective approach.

These findings emphasise the importance of personalised treatment strategies, ensuring that therapy selection is tailored to individual disease characteristics and response patterns. Given that [^177^Lu]Lu-PSMA monotherapy already provides substantial survival benefits, further research should identify which patients would benefit most from combination therapy and which would be optimally treated with monotherapy. The lack of a significant survival advantage with SOC + [^177^Lu]Lu-PSMA suggests that treatment sequencing decisions should be carefully evaluated, as unnecessary combination therapy could introduce additional costs and potential toxicity without meaningful survival gains.

The study results reinforce that radioligand therapy is a powerful treatment option and highlight the need for a strategic, patient-specific approach to maximise its clinical benefits. The clinical implications of these findings strongly support the integration of [^177^Lu]Lu-PSMA into standard treatment protocols for metastatic castration-resistant prostate cancer.

In oncology, the concept of “value” typically encompasses the balance between clinical benefit, side effects, and cost-effectiveness. Frameworks such as the ESMO-Magnitude of Clinical Benefit Scale (ESMO-MCBS) [32] and the ASCO Value Framework [33] provide structured approaches for quantifying value. While these frameworks were not formally applied, our integrated evaluation of survival outcomes and treatment costs reflects their foundational principles.

The Kaplan–Meier AUC-based approach was selected because it enables a dynamic and comprehensive evaluation of clinical and economic benefit across the entire treatment period. Unlike traditional methods based on median survival, this method incorporates the entire survival experience of patients and more accurately reflects the benefit of treatments such as [^177^Lu]Lu-PSMA, which offer durable responses within a predefined number of cycles. In this study, the analysis focused on radiological progression-free survival (rPFS) rather than overall survival (OS), as rPFS more closely aligns with real-world treatment decision-making and cost accrual patterns. Notably, rPFS often represents the clinical juncture at which treatment discontinuation or modification occurs, particularly for fixed-schedule therapies such as [^177^Lu]Lu-PSMA. Conversely, OS does not necessarily correlate with active treatment duration, especially for continuously administered comparators such as ARTA, potentially leading to misalignment between clinical benefit and economic burden.

Although long-term survival extrapolation is often required in pharmacoeconomic models, we adopted a conservative and empirically grounded approach by using reconstructed Kaplan–Meier curves to estimate time-dependent survival through area under the curve (AUC) analysis. This method avoids speculative extrapolations beyond observed follow-up and enables a more reliable estimation of clinical benefit within a fixed treatment window. As a result, our cost evaluations reflect observed efficacy data rather than assumptions about unmeasured long-term outcomes. In this framework, treatment value was interpreted against a standard willingness-to-pay (WTP) threshold of EUR 8000 per month of radiographic progression-free survival (rPFS) gained. Economic comparisons were thus made by calculating the maximum monthly cost that comparators could sustain to remain within acceptable cost–benefit margins relative to [^177^Lu]Lu-PSMA. However, we acknowledge that the lack of extended follow-up data may limit the accuracy of lifetime economic estimates and should be addressed in future studies using formal modelling approaches.

The analysis was performed assuming a cost of EUR 20,000 per dose of [^177^Lu]Lu-PSMA, resulting in a total regimen cost of EUR 120,000 for six fixed administrations. This assumption allows a conservative and realistic estimate of economic impact. However, potential discounts, price negotiations, or national reimbursement agreements could significantly lower acquisition costs, further enhancing the cost-effectiveness of [^177^Lu]Lu-PSMA. These aspects should be considered when evaluating real-world sustainability. 

Assuming a standard WTP threshold of EUR 8,000 per month per rPFS gained, [^177^Lu]Lu-PSMA exhibits a cost-effectiveness profile aligned with accepted sustainability benchmarks, outperforming ARTA and Cabazitaxel in terms of clinical value per euro spent. These break-even considerations underscore the importance of treatment pricing and scheduling in evaluating economic value. Notably, [^177^Lu]Lu-PSMA demonstrates early and consistent rPFS benefits, particularly within the first 18 months, making it a favourable option for patients with rapid disease progression or high disease burden. The bar chart in Figure 7 complements this interpretation by visualising the maximum incremental monthly cost that ARTA and Cabazitaxel could sustain while remaining economically comparable to [^177^Lu]Lu-PSMA under WTP assumptions. This type of break-even threshold analysis reinforces the importance of aligning therapeutic value with affordability.

## 6. Limitations and Future Directions

This study has several limitations. It relies on reconstructed individual patient data from Kaplan–Meier curves, which—despite validation—are subject to approximation and may not fully reflect dynamic survival patterns. Additionally, heterogeneity across trials regarding primary endpoints, treatment sequences, prior exposure, and definitions of standard of care limits direct comparability. For instance, some studies included only taxane-naïve patients or allowed ARTA continuation as SOC, introducing potential bias. Additionally, while this analysis employed break-even cost thresholds anchored to commonly accepted willingness-to-pay (WTP) values, it does not account for country-specific variations in pricing, healthcare reimbursement policies, or societal preferences. Sensitivity analyses exploring different pricing scenarios, time horizons, and survival assumptions will be essential to support more generalisable and equitable decision-making. Finally, identifying predictive biomarkers to select patients most likely to benefit from early [^177^Lu]Lu-PSMA use may further optimise clinical outcomes and economic sustainability.

## 7. Conclusions

This meta-analysis supports a paradigm shift in managing mCRPC, in which radioligand therapy—particularly [^177^Lu]Lu-PSMA—emerges as a front-line option for appropriately selected patients. Combining clinical efficacy, a predictable safety profile, and favourable cost-effectiveness within fixed-cycle regimens, [^177^Lu]Lu-PSMA represents a viable alternative to current standards and may define a new therapeutic benchmark in PSMA-positive disease.

## Figures and Tables

**Figure 1 cancers-17-02247-f001:**
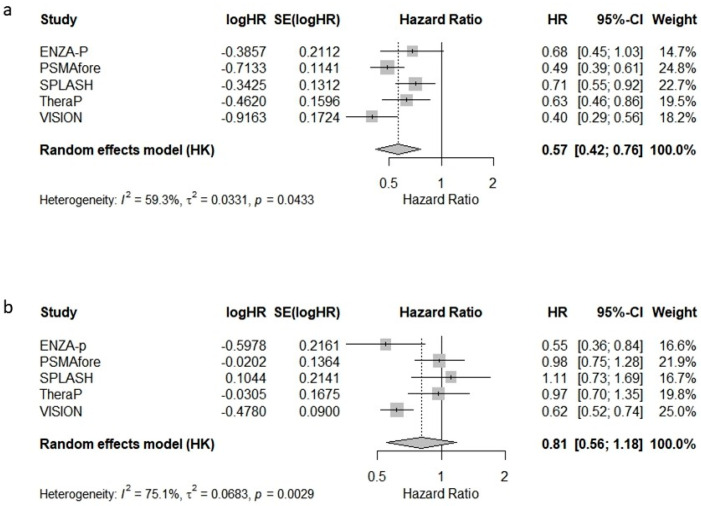
Forest plots of the meta-analysis of hazard ratios (HRs) for (**a**) radiographic progression-free survival (rPFS) and (**b**) overall survival (OS) in patients with metastatic castration-resistant prostate cancer treated with [^177^Lu]Lu-PSMA-617.

**Figure 2 cancers-17-02247-f002:**
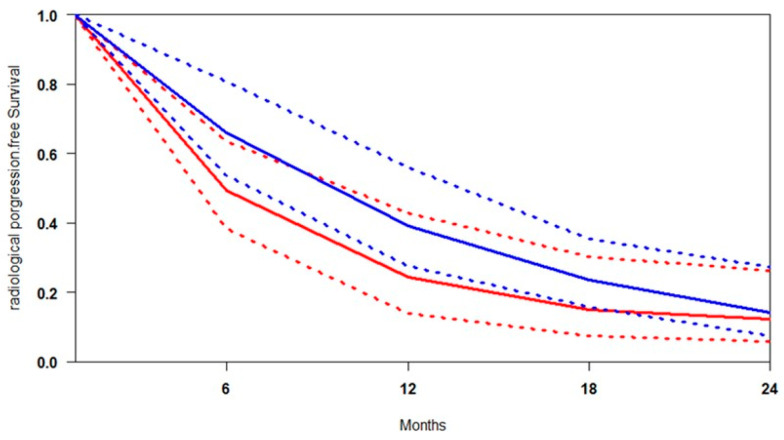
Curves of radiological progression-free survival obtained using the approach of Combescure et al. with random effects. Blue lines represent the summarised radiological progression curves and the 95% confidence bands (dashed lines) for [^177^Lu]Lu-PSMA. Red lines represent the summarised disease progression curves and the control arm’s 95% confidence bands (dashed lines).

**Figure 3 cancers-17-02247-f003:**
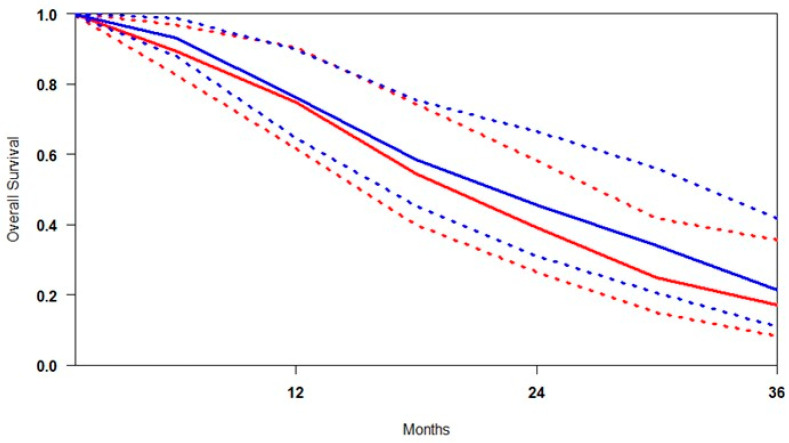
Curves of overall survival obtained using the approach of Combescure et al. with random effects. Blue lines represent the summarised overall survival curves and the 95% confidence bands (dashed lines) for [^177^Lu]Lu-PSMA. Red lines represent the overall survival curves and the control arm’s 95% confidence bands (dashed lines).

**Figure 4 cancers-17-02247-f004:**
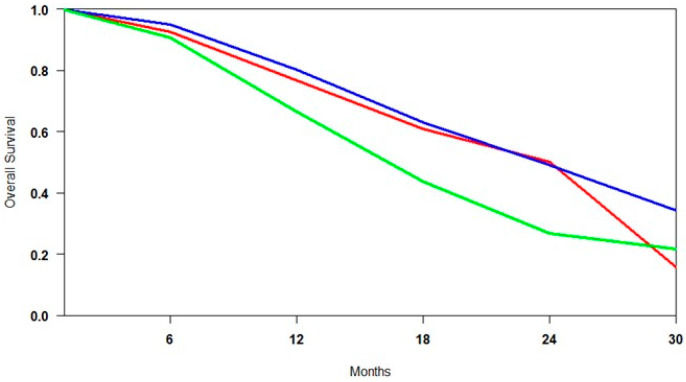
Curves of overall survival of control arms obtained using the approach of Combescure et al. with random effects. Blue lines represent the summarised overall survival curves for ARTA-treated patients. The red lines represent the overall survival curves for patients treated with SOC, while the green lines represent those for patients treated with Cabazitaxel.

**Figure 5 cancers-17-02247-f005:**
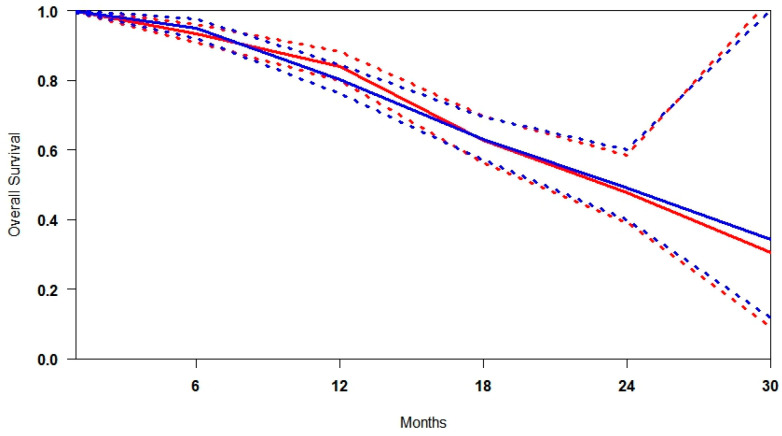
Curves of overall survival obtained using the approach of Combescure et al. with random effects. Blue lines represent the summarised overall survival curves for [^177^Lu]Lu-PSMA. Red lines represent ARTA’s overall survival curves, with 95% confidence bands (dashed lines) shown.

**Figure 6 cancers-17-02247-f006:**
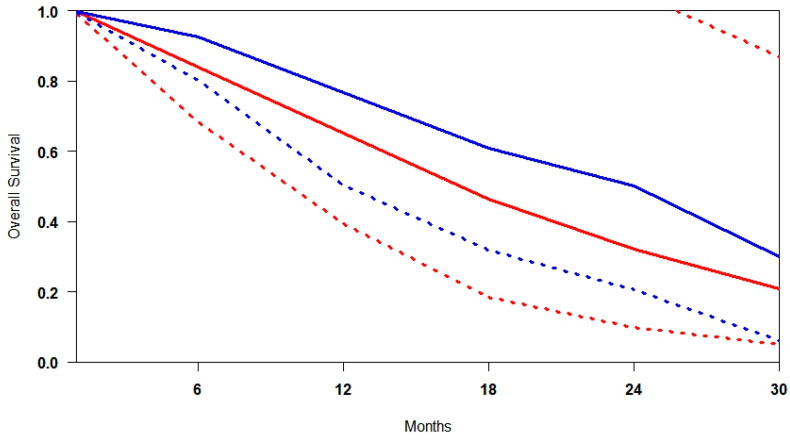
Curves of overall survival obtained using the approach of Combescure et al. with random effects. Blue lines represent the summarised overall survival curves for [^177^Lu]Lu-PSMA+SOC. Red lines represent the overall survival curves, and the 95% confidence bands (dashed lines) are shown for SOC. Confidence intervals extending beyond the upper bound of 1.0 are not displayed for clarity, as this occurs only at early time points with high survival probabilities.

**Figure 7 cancers-17-02247-f007:**
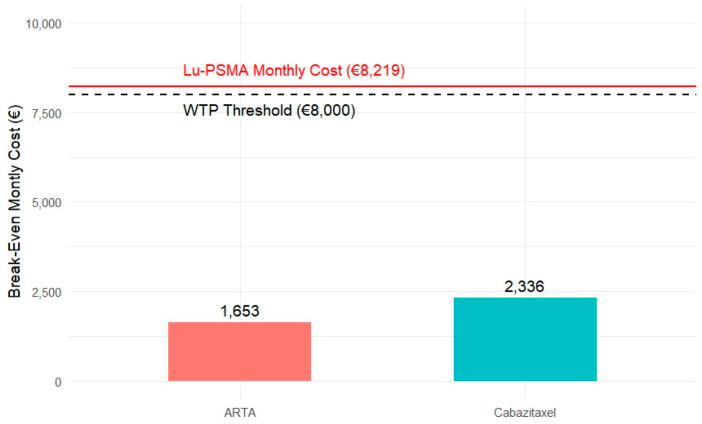
Break-even monthly cost comparison for ARTA and Cabazitaxel about [^177^Lu]Lu-PSMA. The bars represent the maximum incremental monthly cost that ARTA (€1653) and Cabazitaxel (€2336) could sustain while remaining economically comparable to [^177^Lu]Lu-PSMA, based on their respective radiographic progression-free survival (rPFS) gains. The dashed black line indicates the willingness-to-pay (WTP) threshold, while the solid red line represents the estimated monthly cost of [^177^Lu]Lu-PSMA (€8219).

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
