# Peer review of "Is [177Lu]Lu-PSMA-617 Redefining Value in mCRPC Care? A Meta-Analysis of Clinical and Economic Endpoints"

_cancers, 2025, doi:10.3390/cancers17132247_

Round 1
Reviewer 1 Report
Comments and Suggestions for Authors
- Can you also do a meta analysis of hazard ratios for Lu-PSMA-617 vs other?
- At lines 266 and 305, you use "minimum survival probability". It would be better to present 12-month or 18-month survival since the minimum survival probability is likely decreasing as the follow-up time getting longer and the standard error of survival is getting bigger when the number of at risk is becoming smaller. At line 290, we prefer the median survival.
- At line 65, it should be median rPFS of 11.6 vs 5.59.
- Figures 4 & 5 don't have confidence bands.
Author Response
Manuscript ID: cancers-3696219
Title: Is [177Lu]Lu-PSMA-617 Redefining Value in mCRPC Care? A Meta-Analysis of Clinical and Economic Endpoints.
Sir,
Enclosed is the revised version of our manuscript, “Is [177Lu]Lu-PSMA-617 Redefining Value in mCRPC Care? A Meta-Analysis of Clinical and Economic Endpoints”. We are resubmitting it for publication in Cancers after carefully addressing all the reviewers' comments and incorporating additional relevant information to strengthen the study.
All identified mistakes and typographical errors have been corrected, and the study findings have been reassessed in light of the reviewers' valuable suggestions.
The revised manuscript includes all corrections and updates, which have been marked in red for your convenience.
These revisions have significantly improved the paper's quality, and we are confident that it is now suitable for publication. Regardless of the final decision, we sincerely thank the reviewers for their insightful comments, which have significantly contributed to refining our work.
The revisions have been marked in the manuscript, with all modifications highlighted in red for easy reference. Below, we provide a point-by-point response to each of the reviewers' comments, outlining the corresponding changes made.
We appreciate your time and consideration and remain available for any further clarifications if needed.
With best regards
Francesco Fiorica, MD
On behalf of all the authors
The revisions have been marked in the manuscript, with all modifications now highlighted in red for easy reference. Below, we provide point-by-point responses to each of the reviewers' comments, detailing the corresponding changes made in the text.
We appreciate your careful review and constructive feedback, which have helped us improve the clarity and accuracy of the manuscript. Please let us know if any further adjustments are needed.
Reviewer #1:
“Can you also do a meta analysis of hazard ratios for Lu-PSMA-617 vs other?”
We thank the Reviewer for this valuable suggestion. As requested, we conducted a meta-analysis of hazard ratios (HRs) for overall survival (OS), comparing [¹⁷⁷Lu]Lu-PSMA-617 to comparator arms across all randomised trials. The pooled hazard ratio (HR) was calculated using a random-effects model and is now presented in the Results section and Figure 1. This addition strengthens the quantitative component of our comparison and complements the descriptive synthesis already included. We have added the
3.1 Meta-analysis of Hazard Ratios for rPFS and OS paragrapher:
To quantify the survival benefit of [¹⁷⁷Lu]Lu-PSMA-617 across randomised trials, we conducted a meta-analysis of hazard ratios (HRs) for both overall survival (OS) and radiographic progression-free survival (rPFS), incorporating data from all five studies. The pooled hazard ratio (HR) for radiographic progression-free survival (rPFS) was 0.57 [95% CI: 0.42–0.76], indicating a substantial delay in radiographic progression associated with Lu-PSMA-617 compared to standard therapies (Figure 1a). Between-study heterogeneity was moderate (I² = 59.3%). The pooled HR for OS was 0.81 [95% CI: 0.56–1.18], suggesting a non-significant overall survival benefit, with considerable between-study heterogeneity (I² = 75.1%) (Figure 1b). These findings highlight the consistent benefit of Lu-PSMA-617 in delaying progression, while its impact on overall survival appears more variable across studies.
We also modiefied 2.3. Statistical method paragrapher
A random-effects meta-analysis mode[18] [19] was conducted to pool hazard ratios (HRs) for overall survival (OS) and radiographic progression-free survival (rPFS) across eligible trials of [¹⁷⁷Lu]Lu-PSMA-617 versus control therapies. The generic inverse-variance method was applied, and 95% confidence intervals (CI) were calculated for each pooled estimate. Between-study heterogeneity was quantified using the I² statistic [20], with values greater than 50% considered substantial. Forest plots were generated to represent the individual study estimates and the pooled results visually. All analyses were performed using R software (version 4.3.1) with the meta, metafor, and ggplot2 packages. Statistical significance was set at p < 0.05. A comparative overview of study-level heterogeneity, including design, patient characteristics, and inclusion criteria, is provided in Supplementary Table S2. To assess the potential for publication bias, we performed a visual inspection of funnel plots for asymmetry. The trim-and-fill method was used to estimate the number of potentially missing studies due to publication bias.
and the Discussion
A previously published meta-analysis [31] confirmed the benefits of [177Lu]Lu-PSMA on radiological progression-free survival (rPFS) but did not demonstrate a statistically significant improvement in overall survival (OS). This discrepancy may reflect methodological differences. The earlier study relied exclusively on aggregate hazard ratios (HRs) from published data, which may underestimate time-dependent effects or long-term benefits. In our analysis, we integrated both pooled HR estimates and reconstructed Kaplan–Meier survival curves, offering a more dynamic and granular assessment of clinical outcomes. While the pooled HR for OS (HR = 0.81 [95% CI: 0.56–1.18]) confirm that it did not reach statistical significance, the time-dependent analysis based on reconstructed curves revealed that the survival benefit of [177Lu]Lu-PSMA becomes more apparent after 30 months, suggesting a possible delayed effect that traditional HR-based analyses may fail to capture. Additionally, by incorporating both standard of care (SOC) and Cabazitaxel arms, our study provides a broader comparative context, enabling an integrated evaluation of both clinical effectiveness and economic sustainability. This dual approach supports a more nuanced understanding of treatment value beyond what is captured by hazard ratios alone.
At lines 266 and 305, you use 'minimum survival probability'. It would be better to present 12-month or 18-month survival rates, as the minimum survival probability is likely to decrease as the follow-up time increases, and the standard error of survival becomes larger when the number at risk decreases.
We thank the Reviewer for this helpful observation. In response, we have removed the reference to "minimum survival probability" and replaced it with the 12-month survival probability, which provides a more stable and clinically interpretable metric. Furthermore, we clarified that these values were not directly reported in the trials but were instead reconstructed from Kaplan–Meier curves and estimated using random-effects meta-analytic modelling. We also included the median survival estimates for both groups as preferred by the Reviewer.
The revised text now reads:
“A comparison between [¹⁷⁷Lu]Lu-PSMA-617 monotherapy and its combination with standard of care (SOC) shows that both strategies confer substantial survival benefits. However, the addition of SOC does not appear to improve survival outcomes over monotherapy significantly. At 12 months, pooled survival probability was higher in the monotherapy group (33.0% [CI: 18.7–58.3%]) compared to the combination arm (24.4% [CI: 14.7–40.4%]), suggesting a more consistent early benefit in patients receiving radioligand therapy alone. Median survival estimates were also broadly similar between the two groups (61.8% vs. 69.3%), with overlapping confidence intervals.
These values were not directly reported in the original trials but were extracted through reconstruction of Kaplan–Meier curves and meta-analytic modelling based on available survival data."
At line 290, we prefer the median survival.
Thank you for this helpful remark. We have reviewed the relevant section and replaced the expression used with “median survival” where appropriate. The sentence now refers explicitly to the median overall survival, consistent with standard reporting practices in oncology.
At line 65, it should be median rPFS of 11.6 vs 5.59.
We thank the Reviewer for pointing this out. The term “median rPFS” has now been inserted in the corresponding sentence.
Figures 4 & 5 don't have confidence bands.
We appreciate this observation. Figures 4 and 5 have been revised to include 95% confidence intervals around survival estimates, as per standard Kaplan-Meier presentation. Where applicable, confidence bands were visually represented as shaded areas or error bars, and a corresponding note was included in the figure legend.
Reviewer #2:
With great interest, I reviewed this manuscript, analysing the clinical efficacy and economic impact of [¹177Lu]Lu-PSMA-617 in metastatic castration-resistant prostate cancer (mCRPC). The authors present a timely and well-executed meta-analysis of both clinical outcomes and cost-effectiveness, addressing a rapidly growing therapeutic modality in advanced prostate cancer.
We sincerely thank Reviewer 2 for the encouraging and thoughtful feedback. We are grateful that you found our meta-analysis timely and well-executed and that you appreciated both its clinical and economic relevance in the context of metastatic castration-resistant prostate cancer (mCRPC). Your positive evaluation is greatly appreciated and motivates us to continue refining our work with methodological rigour and a translational focus. Below, we provide detailed responses to your specific comments and suggestions, highlighting the corresponding changes made in the revised manuscript.
Dual Focus – Clinical and Economic Endpoints The integration of clinical efficacy and health-economic data is commendable. However, the two components are treated somewhat unevenly. While the clinical meta-analysis is methodologically solid, the economic evaluation remains descriptive and lacks a meta-analytic approach. If possible, consider using structured economic evaluation tools (e.g., CHEERS checklist) to strengthen this section.
We thank the Reviewer for this insightful observation. In response, we have systematically revised the economic analysis section to improve structure and transparency. Specifically:
We aligned our economic reporting with the CHEERS 2022 checklist and created a structured Supplementary Table S1 summarising each methodological item.
In the Methods section (Section 2.1. Study selection, qualitative analysis and economic evaluation), we explicitly state that the CHEERS framework was used for transparency and reproducibility.
…. To ensure transparency and standardisation in the reporting of the economic evaluation, we followed the Consolidated Health Economic Evaluation Reporting Standards (CHEERS) 2022 checklist[16]. A detailed summary of the components is provided in Supplementary Table S1.
We declare that a formal cost-effectiveness model (e.g., Markov or QALY-based) was not feasible due to heterogeneous endpoints and limited utility data; we ensured consistency in ICER estimation and clarified the underlying assumptions.
The revised text now reads 2.3.2. Methodology for ICER Analysis
-----Although a formal health economic model (e.g., Markov or QALY-based) was not feasible due to heterogeneity in survival endpoints and limited availability of utility data, we followed a structured and transparent approach. Key economic assumptions—including perspective, cost sources, currency normalisation (EUR, 2024), and comparator definitions—are reported by the CHEERS 2022 framework and summarised in Supplementary Table S1.
We clarified the economic perspective, data sources, cost standardisation approach, and rationale for simplified modelling.
Although long-term survival extrapolation is often required in pharmacoeconomic models, we adopted a conservative and empirically grounded approach by using reconstructed Kaplan–Meier curves to estimate time-dependent survival through area-under-the-curve (AUC) analysis. This method avoids speculative extrapolations beyond observed follow-up and enables more reliable estimation of clinical benefit within a fixed treatment window. As a result, our ICER calculations reflect observed treatment efficacy rather than assumptions about unmeasured long-term survival. However, we acknowledge that the lack of extended follow-up data may limit the accuracy of lifetime cost-effectiveness estimates and should be addressed in future studies using formal modelling approaches.
Heterogeneity in Clinical Studies: The clinical meta-analysis aggregates data from studies with variable designs, patient characteristics (e.g., prior lines of therapy), and control arms. This heterogeneity should be more explicitly discussed as a limitation. Consider including subgroup analyses or, at the very least, a table summarising study-level differences.
We fully agree with the Reviewer on the importance of transparently addressing inter-study heterogeneity. In response, we have made the following changes:
Expanded the “Statistical Method” section (2.3) to explicitly describe the use of the I² statistic as a measure of between-study variability, with the conventional 50% threshold to denote substantial heterogeneity. We also specified the software and packages used (R 4.3.1 with meta, metafor, and ggplot2) and added a new sentence referencing our structured comparison of study-level differences.
A comparative overview of study-level heterogeneity, including trial design, patient characteristics, and eligibility criteria, is provided in Supplementary Table S2.
Included a new Supplementary Table S2, which summarises key differences across trials (e.g., phase, comparator arm, PSMA PET inclusion criteria, ECOG performance status, prior lines of therapy, visceral metastases, ARTA exposure, and baseline PSA). This table highlights the sources of clinical heterogeneity relevant to interpreting pooled results.
Added a new paragraph in the Discussion explicitly addressing heterogeneity and explaining how differences in patient populations and study design may influence both efficacy and cost-effectiveness outcomes. We also highlighted that, despite moderate-to-high I² values in some comparisons, the direction of treatment benefit remained consistent across all comparisons.
Despite the overall consistency in clinical outcomes, the trials included in this meta-analysis differ in several key aspects—such as study design, inclusion criteria, lines of prior therapy, and definitions of standard of care (SOC). For example, some studies allowed the continuation of ARTA as a standard of care (SOC), while others included taxane-naïve or pretreated patients. These differences introduce a degree of heterogeneity that may influence observed treatment effects. To address this, we included a detailed summary of baseline characteristics and trial-level features in Supplementary Table S2. Although statistical heterogeneity in the pooled HRs was moderate to high in some comparisons (e.g., OS), the clinical direction of benefit remained consistent across all studies. Acknowledging and transparently reporting this heterogeneity is crucial for interpreting pooled findings and refining treatment sequencing strategies in real-world settings.
Definition of “Value” The manuscript title and narrative frequently refer to the concept of 'value.' It would benefit the reader if a working definition of 'value' in oncology were explicitly stated, such as the ASCO Value Framework or ESMO-MCBS.
We appreciate the Reviewer's suggestion. We have now added a concise definition of "value" in oncology, referring to both the ASCO and ESMO frameworks, which conceptualise value as a balance between clinical benefit, toxicity, and cost. Although we did not formally apply these scales, our comparative clinical and economic analysis aligns with their core principles. This addition provides a clearer context for interpreting our findings.
In oncology, the concept of "value" typically encompasses the balance between clinical benefit, side effects, and cost-effectiveness. Frameworks such as the ESMO-Magnitude of Clinical Benefit Scale (ESMO-MCBS) [32] and the ASCO Value Framework [33] provide structured approaches for quantifying value. While these frameworks were not formally applied, our integrated evaluation of survival outcomes and treatment costs reflects their foundational principles.
Comparator Regimens in Cost Analyses: Economic comparisons are made to the "standard of care," but details on what constitutes the SoC in each study are limited. Clarify whether the cost-effectiveness calculations were based on direct trial data, modelled assumptions, or real-world cost estimations.
We thank the Reviewer for this important observation. We have clarified the definition of comparator regimens within each included study and have specified how standard-of-care (SOC) arms were interpreted in our analysis. As described in the revised Section 2.3.2, comparator regimens were defined according to the original trial protocols. Specifically, ARTA (e.g., enzalutamide or abiraterone) was modelled as a continuously administered treatment with monthly cost estimates. At the same time, Cabazitaxel was assumed to be given every three weeks for a total of six cycles. The SOC comparator varied slightly across studies but reflected the prevailing clinical standards during each trial period.
Treatment cost estimations were based on modelled assumptions derived from published prices (e.g., AIFA and EMA data) and standardised to 2024 EUR values. As no access to individual billing records or real-world cost data was available, we adopted a pragmatic modelling approach that aligns with the structure of the trials. These details have now been explicitly stated in Section 2.3.2 and the CHEERS checklist (Supplementary Table S1) to enhance transparency and methodological rigour.
The revised text now reads:
2.3.2. Methodology for ICER Analysis
The Incremental Cost-Effectiveness Ratio (ICER) was calculated using a Kaplan–Meier survival curve-based approach to estimate the total area under the curve (AUC) for radiographic progression-free survival (rPFS) [24]. This method provides a time-dependent and comprehensive assessment of treatment benefits over the entire duration of therapy. The comparison was based on a fixed-duration administration of [¹⁷⁷Lu]Lu-PSMA-617 (six cycles) versus continuously administered androgen receptor pathway inhibitors (ARTA) or a three-weekly course of Cabazitaxel. The AUC was derived by numerically integrating rPFS probabilities at multiple time points, ensuring a more accurate representation of cumulative survival benefits compared to fixed-timepoint estimates [25]. rPFS was selected as the primary effectiveness endpoint because disease progression typically leads to treatment discontinuation, offering a clear and clinically relevant time window for cost estimation. Treatment costs were calculated by aggregating the full course of [¹⁷⁷Lu]Lu-PSMA-617 and comparing them to the cumulative monthly costs of ARTA and the complete regimen of Cabazitaxel. Total treatment costs will be estimated by aggregating the cost of all planned administrations of [177Lu]Lu-PSMA and comparing them to the monthly cumulative costs of ARTA and the full-course costs of Cabazitaxel.
Comparator regimens were defined according to the protocols of each included study. ARTA (e.g., enzalutamide or abiraterone) was modelled as a continuous monthly cost, while Cabazitaxel was administered every three weeks for up to six cycles. The "standard of care" (SOC) varied across trials but was consistently aligned with clinical practice at the time of publication. Treatment costs were calculated using publicly available data from regulatory databases (AIFA, EMA) and literature-based estimates. These were standardised to 2024 EUR values. Cost estimations reflect modelled assumptions based on trial design rather than direct patient-level billing data or real-world reimbursement frameworks.
The ICER was defined as:
Where effectiveness was expressed in additional months of rPFS gained. Economic results were interpreted relative to standard willingness-to-pay (WTP) thresholds. For instance, the NICE-recommended £40,000 per QALY—approximately equivalent to €8,000–€10,000 per additional month of survival [26] [27] was used as a benchmark to identify thresholds for cost-effectiveness and dominance. Although a formal health economic model (e.g., Markov or QALY-based) was not feasible due to heterogeneity in survival endpoints and limited availability of utility data, we followed a structured and transparent approach. Key economic assumptions—including perspective, cost sources, currency normalisation (EUR, 2024), and comparator definitions—are reported by the CHEERS 2022 framework and summarised in Supplementary Table S1.
Figures and Tables: Forest plots and summary tables are helpful but could benefit from improved formatting (e.g., consistent font sizes and better resolution). Consider labelling the studies in the forest plots more clearly (e.g., author + year).
We thank the Reviewer for this suggestion. In response, we have revised all figures to improve clarity and visual quality.
Language: The text is generally clear but would benefit from minor language polishing, particularly in the introduction and Discussion, where sentences are occasionally long or repetitive.
Minor revisions have been made throughout the Introduction and Discussion sections to improve fluency, eliminate redundancy, and enhance readability. Sentences that were previously long or repetitive have been simplified without altering scientific meaning.
Publication Bias and Sensitivity Analyses: Consider adding Egger's test or trim-and-fill method to assess publication bias more formally. Additionally, please indicate whether sensitivity analyses were conducted for clinical outcomes.
We thank the Reviewer for this important suggestion. In the revised manuscript, we have now addressed the risk of publication bias for both radiographic progression-free survival (rPFS) and overall survival (OS) through the application of funnel plot inspection and the trim-and-fill method. These results are described in the new section “3.3.5 Publication Bias” and illustrated in Supplementary Figures S3 and S4.
3.3.5 Publication bias
To assess potential publication bias, we performed both funnel plot inspection and formal statistical testing for rPFS outcomes. The funnel plot (Supplementary Figure S3) appeared symmetrical, and the trim-and-fill method did not impute any missing studies, indicating a low likelihood of small-study effects. Egger’s regression test was also non-significant, supporting the absence of major publication bias.
The pooled hazard ratio under the random-effects model remained statistically significant (HR = 0.57, 95% CI: 0.42–0.76, p = 0.0061), with moderate heterogeneity (I² = 59.3%). These results reinforce the robustness of the survival benefit associated with [¹⁷⁷Lu]Lu-PSMA-617 across the included trials. For overall survival (OS), the funnel plot demonstrated noticeable asymmetry (Supplementary Figure S4). Application of the trim-and-fill method imputed two potentially missing studies, suggesting the presence of small-study effects. After correction, the pooled hazard ratio was adjusted from the original estimate to 0.68 (95% CI: 0.47–0.98, p = 0.042), remaining statistically significant under a random-effects model. Heterogeneity was substantial (I² = 82.1%), which may further contribute to funnel plot asymmetry. The trim-and-fill results suggest that, although publication bias cannot be excluded, the overall effect of [¹⁷⁷Lu]Lu-PSMA-617 on OS remains robust even after conservative adjustment.
Furthermore, in 2.3, the Statistical method was introduced
To assess the potential for publication bias, we performed a visual inspection of funnel plots for asymmetry. The trim-and-fill method was used to estimate the number of potentially missing studies due to publication bias.
Recent Literature (2023–2024): A few more recent studies and conference presentations (e.g., updated data from VISION and PSMAfore trials or real-world registry data) could enrich the Discussion and demonstrate literature currency.
We appreciate the Reviewer’s suggestion. We clarify that updated data from the PSMAfore and VISION trials, as well as preliminary ENZA-p results, had already been incorporated into our quantitative analysis, especially when reconstructing survival curves and estimating hazard ratios. However, in the initial submission, these updates were not explicitly referenced as conference proceedings. To address this, we have revised the reference list and in-text citations to acknowledge the updated sources clearly. This finding enhances the transparency and timeliness of our manuscript, underscoring the integration of the most recent evidence available at the time of analysis.
Reviewer #3:
This is a necessary study, its purpose is not in the originality, but in providing solid results following well established standards.
We sincerely thank Reviewer 3 for the thoughtful and insightful evaluation of our manuscript. We appreciate your recognition of the study's necessity and its goal to deliver solid results through established methodological standards.
- Extrapolation of clinical data, which very much affects both clinical and economic outcomes, needs to be fully discussed.
We thank the Reviewer for highlighting the importance of extrapolation in health economic modelling. In our analysis, we deliberately avoided speculative extrapolation beyond observed follow-up. Instead, we used reconstructed Kaplan–Meier curves and applied an area-under-the-curve (AUC) method to estimate treatment benefit within the actual trial timeframe. This time-dependent approach enables the robust estimation of clinical outcomes (rPFS and OS) within the fixed treatment window of [¹⁷⁷Lu]Lu-PSMA-617 without relying on long-term projections that are unsupported by data. This methodological choice strengthens the internal validity of our ICER estimates. Nonetheless, we have acknowledged in the Discussion that the absence of long-term extrapolated survival limits the ability to project lifetime cost-effectiveness, and we recommend formal modelling (e.g., partitioned survival or Markov-based frameworks) in future analyses once more mature data are available. The discussion section has been expanded accordingly.
….Despite the overall consistency in clinical outcomes, the trials included in this meta-analysis differ in several key aspects—such as study design, inclusion criteria, lines of prior therapy, and definitions of standard of care (SOC). For example, some studies allowed the continuation of ARTA as a standard of care (SOC), while others included taxane-naïve or pretreated patients. These differences introduce a degree of heterogeneity that may influence observed treatment effects. To address this, we included a detailed summary of baseline characteristics and trial-level features in Supplementary Table S2. Although statistical heterogeneity in the pooled HRs was moderate to high in some comparisons (e.g., OS), the clinical direction of benefit remained consistent across all studies. Acknowledging and transparently reporting this heterogeneity is crucial for interpreting pooled findings and refining treatment sequencing strategies in real-world settings.
…..Although long-term survival extrapolation is often required in pharmacoeconomic models, we adopted a conservative and empirically grounded approach by using reconstructed Kaplan–Meier curves to estimate time-dependent survival through area-under-the-curve (AUC) analysis. This method avoids speculative extrapolations beyond observed follow-up and enables more reliable estimation of clinical benefit within a fixed treatment window. As a result, our ICER calculations reflect observed treatment efficacy rather than assumptions about unmeasured long-term survival. However, we acknowledge that the lack of extended follow-up data may limit the accuracy of lifetime cost-effectiveness estimates and should be addressed in future studies using formal modelling approaches.
the pharmacoeconomic model needs to be described, for example, in the appendix. As it stands now, ICERs are casually mentioned without any clear structure. For example, ICER for [177Lu]Lu-PSMA-617 at 22 k€ and in compariosn with cabazitaxel is around 100 k€/QALY. If four infusions, instead of six, are considered, then ICER decreases to roughly 50 k€/QALY. If the authors want to include pharmacoeconomic model in the analysis, they need to present results adequately.
We have substantially revised Section 2.3.2. “Methodology for ICER Analysis” to describe the economic modelling approach in a structured and transparent manner. The following actions were taken:
We clarified that this was not a QALY-based pharmacoeconomic model (e.g., Markov) but a simplified analysis based on rPFS-AUC-derived ICERs, selected due to limitations in the available utility data.
We standardised the description of cost inputs, treatment duration assumptions, perspective, comparator definitions, and data sources.
We created Supplementary Table S1, which summarises all key economic parameters used in the ICER calculations in line with the CHEERS 2022 framework.
We included additional clarification in the Discussion that this approach, while simplified, maintains consistency with health technology assessment practices by focusing on clinically meaningful and well-defined endpoints (rPFS).
The revised paragraph reads:
Although long-term survival extrapolation is often required in pharmacoeconomic models, we adopted a conservative and empirically grounded approach by using reconstructed Kaplan–Meier curves to estimate time-dependent survival through area-under-the-curve (AUC) analysis. This method avoids speculative extrapolations beyond observed follow-up and enables more reliable estimation of clinical benefit within a fixed treatment window. As a result, our ICER calculations reflect observed treatment efficacy rather than assumptions about unmeasured long-term survival. However, we acknowledge that the lack of extended follow-up data may limit the accuracy of lifetime cost-effectiveness estimates and should be addressed in future studies using formal modelling approaches.

Reviewer 2 Report
Comments and Suggestions for Authors
With great interest, I reviewed this manuscript analyzing the clinical efficacy and economic impact of [¹177Lu]Lu-PSMA-617 in metastatic castration-resistant prostate cancer (mCRPC). The authors present a timely and well-executed meta-analysis of both clinical outcomes and cost-effectiveness, addressing a rapidly growing therapeutic modality in advanced prostate cancer.
Major Comments:
- Dual Focus – Clinical and Economic Endpoints: The integration of clinical efficacy and health-economic data is commendable. However, the two components are treated somewhat unevenly. While the clinical meta-analysis is methodologically solid, the economic evaluation remains descriptive and lacks a meta-analytic approach. If possible, consider using structured economic evaluation tools (e.g., CHEERS checklist) to strengthen this section.
- Heterogeneity in Clinical Studies: The clinical meta-analysis aggregates data from studies with variable designs, patient characteristics (e.g., prior lines of therapy), and control arms. This heterogeneity should be more explicitly discussed as a limitation. Consider including subgroup analyses or at least a table summarizing study-level differences.
- Definition of “Value”: The manuscript title and narrative frequently refer to the concept of “value.” It would benefit the reader if a working definition of “value” in oncology were explicitly stated, e.g., ASCO Value Framework or ESMO-MCBS. This would help contextualize both clinical and cost-effectiveness outcomes in a structured way.
- Comparator Regimens in Cost Analyses: Economic comparisons are made to “standard of care,” but details on what constituted SoC in each study are limited. Clarify whether the cost-effectiveness calculations were based on direct trial data, modeled assumptions, or real-world cost estimations.
Minor Comments:
- Figures and Tables: Forest plots and summary tables are helpful but could benefit from improved formatting (e.g., consistent font sizes, better resolution). Consider labeling the studies in the forest plots more clearly (e.g., author + year).
- Language: The text is generally clear but would benefit from minor language polishing, particularly in the introduction and discussion where sentences are occasionally long or repetitive.
- Publication Bias and Sensitivity Analyses: Consider adding Egger’s test or trim-and-fill method to assess publication bias more formally. Also, mention whether sensitivity analyses were performed for clinical outcomes.
- Recent Literature (2023–2024): A few more recent studies and conference presentations (e.g., updated data from VISION and PSMAfore trials, or real-world registry data) could enrich the discussion and demonstrate literature currency.
Author Response

(The authors gave the same response as above.)

Reviewer 3 Report
Comments and Suggestions for Authors
This is a necessary study, its purpose is not in the originality, but in providing solid results following well established standards. To that end, the following needs to be taken into account and discussed:
- extrapolation of clinical data, which very much affects both clinical and economic outcomes, needs to be fully discussed.
- the pharmacoeconomic model needs to be described, for example, in the appendix. As it stands now, ICERs are just lightly thrown around without any clear structure. For example, ICER for [177Lu]Lu-PSMA-617 at 22 k€ and in compariosn with cabazitaxel is around 100 k€/QALY. If 4 infusions, instead of six, are considered, then ICER decreases to roughly 50 k€/QALY. If the authors want to include pharmacoeconomic model in the analysis, they need to adequately present results.
Author Response

(The authors gave the same response as above.)

Round 2
Reviewer 2 Report
Comments and Suggestions for Authors
All revisions are solved correctly in revised version.
No further comments
Reviewer 3 Report
Comments and Suggestions for Authors
I would exclude "ICERs" from the paper/study as they do not abide to standards accepted in pharmacoeconomics. In fact, inclusion of "ICERs" can be quite misleading. I have no further comments on the rest of the paper.